



# The European forest Carbon budget under future climate conditions and current management practices

Roberto Pilli[1], Romain Alkama[2], Alessandro Cescatti[2], Werner A. Kurz[3], Giacomo Grassi[2*]

[1] Joint Research Centre (JRC) Consultant, 35133 Padova (PD), Italy

[2] European Commission, Joint Research Centre, Directorate D – Sustainable Resources – Bio-Economy Unit, Via E. Fermi 2749, I-21027 Ispra (VA), Italy

[3] Natural Resources Canada, Canadian Forest Service, Victoria, BC, V8Z 1M5, Canada

*Correspondence to*: Giacomo Grassi (giacomo.grassi@ec.europa.eu





**Abstract.** To become carbon neutral by 2050, the European Union (EU27) net carbon sink from forests should increase from the current level of about -360 Mt $CO_{2e}$ yr$^{-1}$ to -450 Mt $CO_{2e}$ yr$^{-1}$ by 2050. Reaching this target requires additional efforts, which should be based on a strategic view of the realistic evolution of European forests within the next decades, considering the current age-class distributions, the effect of forest management practices and the expected impacts of future climate change. However, modelling the combined effect of these drivers is quite

challenging since it requires a mechanistic assessment of climate impacts on primary productivity and a detailed representation of the forest age structure and of the management practices across the entire EU. To achieve this goal, in this study we combined the output provided by four land-climate models - run under two different representative concentration pathway scenarios (RCP 2.6 and RCP 6.0) - to parameterize the input data used in an empirical forest growth model. This hybrid modelling approach aims to quantify the impact of climate change and forest management

on the long-term (i.e., to 2100) evolution of the EU27+UK forest carbon budget, under a Business-as-Usual scenario, based on the continuation of the management practices applied by EU member states within the historical period 2000 - 2015.

Our results highlight that, under the continuation of the current management practices, the EU27+UK forest C sink would decrease to about -250 Mt $CO_{2e}$ yr$^{-1}$ in 2050 and -80 Mt $CO_{2e}$ yr$^{-1}$ by 2100. The main driver of the long-term

evolution of the forest C sink is the ongoing ageing process of the European forests, mostly determined by past management. In addition, climate change may further amplify or mitigate this trend. Due to the large uncertainty of climate projections, in 2050 the net C sink may range from -100 to -400 Mt $CO_{2e}$ yr$^{-1}$ $CO_{2e}$ yr$^{-1}$ (RCP 2.6) and from -100 to -300 Mt $CO_{2e}$ yr$^{-1}$ (RCP 6.0). This suggests that, while a change in management practices is needed to reverse an otherwise declining trend in the sink, climate change adds a considerable uncertainty, potentially nearly doubling

or halving the sink associated to management.

**Keywords**: Climate change, Forest Management, Modelling, Net Biomass Production, EU Forest Strategy



## 1. Introduction


The key role of forests to meet the Paris Agreement's climate targets is widely recognized by the scientific community (IPCC, 2019). This is also relevant for major industrialized countries, where the carbon uptake by forests, including preserving or strengthening the carbon sink and the use of wood to substitute other emissions-intensive materials, will be crucial to compensate any remaining emission from industrial and agricultural sectors (Dugan et al., 2021). To

become carbon neutral by 2050, on top of a drastic decarbonization of energy, transport and industrial sectors, the European Union (EU27) net sink from forest land should increase to about -450 Mt $CO_{2e}$ $yr^{-1}$ by 2050 (EC, 2020a). Considering the recent evolution of this sink – declining from about -410 Mt $CO_{2eq}$ in the period 2010-2012[1] to about -360 Mt $CO_{2eq}$ in 2016-2018 – a new regulation for the Land Use, Land-Use Change and Forestry (LULUCF) sector has been proposed (EC, 2021b) to stimulate additional efforts for reversing the current trend. The emerging debate on

the role of forests in climate change mitigation requires comprehensive analyses on the expected evolution of the forest sink over the next decades (Verkerk et al., 2020).

The short-term evolution of the forest C sink is directly determined by forest management practices and stochastic natural disturbances, which determine forest composition and age structure (Pilli et al., 2016). However, assessing the impact of forest management practices is challenging because of the uncertainties linked to policy and economic

drivers, which directly affect the future harvest rate (see Grassi et al., 2018). For this reason, various studies based on empirical, forest stand growth models may provide different, and sometimes opposite results (Skytt et al. 2021). For example, Nabuurs et al. (2017) estimated that the EU28 forest C sink could potentially increase by about 172 Mt $CO_2$ $yr^{-1}$ by 2050. In contrast, Jonsson et al. (2021), based on different methodological assumptions and harvest scenarios, estimated a reduction of the EU 28 forest C sink by 50 and 180 Mt $CO_2$ $yr^{-1}$ in 2030, compared to 2015. Even assuming

the same harvest level - e.g. a business-as-usual (BaU) scenario based on constant harvest - similar models may produce different results, because of different assumptions on the management strategies applied at the local level, which may, in turn, also affect the long-term evolution of the age class distribution (Blujdea et al. 2021). Therefore, determining a common, possibly "neutral", management scenario, that represents a benchmark for the development of further management strategies, is also challenging (Pukkala, 2020).

When moving to the long-term evolution of the forest C sink, we also need to consider the scientific uncertainties about the evolution of environmental drivers (i.e., temperature, precipitation and atmospheric CO2 concentration) and their impact on the future forest growth (including, for example, species composition and frequency of natural disturbances), and the increasing expectations placed on forests by the ongoing EU policy initiatives (Mubareka et al. 2022). These include not only the climatic policy, where wood removals are part of a climate neutral bioeconomy, but

also the EU biodiversity strategy, where the so call old-growth forests have a key role (EC, 2020b).

Modelling all these drivers is clearly challenging. For modelling the medium- to short-term evolution of these variables, empirical, forest stand growth models are generally best suited (Nabuurs et al. 2000, Böttcher et al. 2008). These models, however, by simulating the forest growth based on past observations, cannot easily determine the potential variations in primary productivity induced by climate changes (Cuddington et al., 2013). On the other hand,

---

[1] Excluding Harvested Wood Products (HWP)



modelling the long-term evolution of the forest C sink to identify large-scale management strategies under climate change conditions, generally requires the use of process-based climate models, grounded in ecological theories. These models, however, generally miss detailed information on management practices and forest conditions, as determined from direct field measurements (Pretzsch et al. 2008).

A compromise solution is to build a meta-modelling framework that merges the strategic information provided by

process-based models, with the accuracy provided by empirical models (Cuddington et al., 2013).

Here we aim to investigate the medium to long-term (i.e., 2050 and beyond) evolution of the forest C sink, as affected by the complex interactions between climatic variables and forest ecosystems. Due to the uncertainty about the future evolution of environmental variables and the relative impact of these variables on forest growth and mortality, we determine a range of outcomes by combing different climatic scenarios and process-based models. The main objective

of our study is to quantify the EU carbon sink dynamics as affected by climate change, forest management and disturbances. To achieve this, we down-scale the output provided by a process-based modelling framework to the empirical growth functions and management practices applied by a stand-level forest growth model. This meta-modelling approach uses state of the art modelling tools, to analyze the combined impacts of climate change and forest management on the long-term (i.e., to 2100) evolution of the forest carbon budget of the EU (hereafter including

EU27+UK), under a scenario of continuation of the current management practices.

## 2. Materials and Methods

### 2.1. Modelling framework

The modelling framework used in this study, summarized on Fig 1S (see Supplementary Materials), integrates statistics of land carbon fluxes from the Inter-Sectoral Impact Model Intercomparison Project (ISI–MIP2b,

Warszawski et al. 2013), which combines dynamic vegetation models (DGVM) and process-based climate models, in the parameterization of the empirical forest growth model used within a forest carbon budget model (CBM-CFS3, Carbon Budget Model, Kurz et al., 2009). Specifically, we used outputs from LPJ-GUESS DGVM (Smith et al. 2014), which is the only model in ISI-MIP that provides all the required variables (i.e. forest net growth and frequency of fires). This DGVM model is forced by four different climate variables (2m air temperature, precipitation, incoming

solar radiation, incoming longwave radiation, surface wind and humidity) coming from four different process-based climate models (IPSL-CM5, Institut Pierre Simon Laplace (Dufresne et al. 2013); GFDL, Global Atmosphere and Land Model (Zhao et al., 2018); HadGEM2, Hadley Centre Global Environmental Model version 2 (Collins et al., 2011); MIROC 5.2, Model for Interdisciplinary Research on Climate (Kawamiya et al., 2020)), which were run under the Coupled Model Intercomparison Project 5 (CMIP5). In the ISI-MIP framework, this climate forcing is interpolated

to a $0.5° \times 0.5°$ spatial resolution and then bias-corrected to ensure long-term statistical agreement with the observation-based forcing data (Warszawski et al. 2013). This combination between climate simulations and LPJ-GUESS DGVM from ISI-MIP2b was used to predict the annual variation of net forest growth and frequency of fires, in the period 2016 - 2100, compared to the historical period 2000 – 2015, assumed as reference period within this overall modelling framework. The net forest growth was estimated as the annual change in the total carbon in


vegetation biomass. Losses from fires are included in the DGVM simulations but not harvest, which is included in the CBM simulations. Each simulation was run under two different representative greenhouse gases concentration pathways scenarios: RCP 2.6 and RCP 6.0, as defined by Taylor et al. 2012 and includes $CO_2$ fertilization effect.

As in other studies (see for example Sun and Mu, 2014), to explore the impact of climate change on forest ecosystems, we combined the carbon in vegetation (cveg) simulated at plant functional types level (PFTs) from LPJ-GUESS, with

the forest types (FTs) considered by CBM, distinguished between broadleaved and coniferous groups. This aggregation is made according to the spatial distribution of the Climatic Units (CLU, defined from specific values of MAT and total annual precipitation) considered within the CBM model (Pilli et. al., 2018). Using this approach, the DGVM input (MAT) and output (cveg as proxy of net growth, and fire area as proxy of frequency of fires) can be directly and consistently integrated with the forest growth model.

In particular, the MAT of each CLU was assumed as constant until 2015 (equal to the average of the MATs' values of the historical period as considered from each climate model) and as varying - compared to the average of the historical period – by year from 2016 onward. This variable affects the decay rates of dead organic matter (DOM) within the CBM model run (see Fig. 3S and 4S in Supplementary Materials).

Based on the annual biomass carbon stock per ha estimated from each LPJ-GUESS simulation, we estimated the

relative annual stock change from 2016 onward, compared to the average of the historical period. We derived from this parameter a set of growth multipliers (GMs, further distinguished between broadleaved species and conifers and scaled at CLU level) for each country directly proportional to the relative variation of the biomass stock as estimated from LPJ-GUESS under each RCP. Starting from 2016, these GMs were applied in CBM to the species-specific growth functions derived from National Forest Inventory (NFI) increment data. In this way, the relative net growth of

each FT in CBM varies according to the impact of climatic conditions as predicted by LPJ-GUESS. In a few cases, where data from process-based models were missing (e.g., for coniferous species in Portugal and Ireland), the GMs were derived from other conterminous regions with similar climatic conditions (see Fig. 5S and 6S in Supplementary Materials).

$CO_2$ emissions due to fires provided from LPJ-GUESS and further scaled at the CLU level, were used as a proxy to

estimate the relative variation of burned area considered by CBM from 2016 onward for 6 Mediterranean countries, in comparison to the average burned area of the historical period (see Fig. 7S in Supplementary Materials). Other natural disturbances, such as windstorms and bark beetle outbreaks, were not accounted in this analysis, because the current modelling framework is still rather uncertain in representing the rates of change of these disturbance types under climate change.

The CBM model was preliminarily calibrated according to the annual harvest rate reported for the historical period 2000 - 2015 from each EU27 member state, plus UK and excluding Malta and Cyprus, where no detailed data are available. All FTs considered by CBM were spatially distributed between 35 CLUs and assigned to broadleaved or coniferous groups, according to the leading species reported from countries' NFI data. The calibration was performed at country level, defining a set of species-specific silvicultural treatments applied to each FT (i.e., thinning and clear-

cut for even-aged forests, partial-cut for uneven-aged forests, etc.) in order to satisfy the historical harvest demand as defined for each country.



### 2.2. Defining a Business-as-Usual scenario for forest management

For forest management, defining a business-as-usual scenario means assuming the continuation, beyond ten to fifteen years into the future, of the current management practices and policies. To achieve this objective, we used the same

approach proposed for the definition of the Forest Reference Level within the Regulation EU 2018/841 (Vizzarri et al., 2021). This approach can be considered as a "business-as-usual" continuation of the forest management practices documented within a certain period of time, defined as reference period (RP, see Grassi et al. 2018). This approach is based on a country-specific assessment of specific forest management practices, characteristics and age-related dynamics. As such, it can inherently incorporate the impact of polices and markets enhanced during the RP, excluding

at the same time, any additional assumption on the possible impact of future polices and markets scenarios (Grassi et al, 2018).

In the present study, after the calibration stage, we quantified for each country the intensity (in terms of proportion of available biomass harvested for each FT and management type, see Grassi et al. 2018) of each management practice applied during the historical period 2000 – 2015 (assumed as RP for our study). The same intensity was applied within

the following simulation period 2016 – 2100 (see Grassi et al., 2018). This is, in turn, was determined from the evolution of the age-class distribution, linked to the natural aging process of forests and from the specific management practices applied at the country level. Thus, the absolute amount of harvest applied within the simulation period 2016-2100 was not preliminarily defined as a constant amount of biomass removed from each country, but it may vary according to the evolution of the age-class distribution within each simulation.

The CBM model was run, for each country, according to the output provided from each LPJ-GUESS simulation, for both RCP scenarios (i.e., eight climate runs per country). One additional model run was simulated as Reference scenario (RS) to compare the LPJ-GUESS outputs with a benchmark, excluding any additional effect of climate change (see Fig. 1S).

### 2.3. Ecosystem indicators

To quantify the combined effect of climatic impacts and management activities on forest ecosystems, we identified a series of key variables derived from the CBM model runs. From an ecosystem perspective, the yearly sum of all biomass production is estimated as Net Primary Production (NPP), equal to the difference between the carbon assimilated by plants through photosynthesis (i.e., the Gross Primary Production, GPP) and the carbon released by plants through autotrophic respiration (Kirschbaum et al., 2001). CBM does not quantify photosynthesis and

autotrophic respiration, but the model indirectly estimates the NPP as the sum of net growth (NG), which is the net biomass increment before losses from disturbances (i.e., it is a measure of biomass C stock change) plus the biomass turnover (TO), i.e., the growth that replaces material lost due to biomass turnover, during the year (Kurz et al., 2009):

$$NPP = NG + TO \qquad \text{Eq. (1)}$$

The Net Ecosystem Production (NEP) is defined as the difference between GPP and the total ecosystem respiration

(Chapin et al., 2006). The CBM estimates NEP by subtracting from NPP all the C losses due to the heterotrophic respiration (Rh, i.e., decomposition):

$$NEP = NPP - R_h \qquad \text{Eq. (2)}$$



In the modelling framework applied in the present study, both growth rates and decomposition rates are modified
during the model run, to account for the effects of climate change.

The overall ecosystem C balance is the Net Biome Production (*NBP*), which is the difference between NEP and the
direct losses due to harvest (*H*) and natural disturbances (*D*, e.g., fires):

$$NBP = NEP - H - D \qquad \text{Eq. (3)}$$

The CBM calculates NBP as the total ecosystem stock change, estimated in annual time steps, as gains from net growth
increment, and losses from the ecosystem due to decay, direct atmospheric emissions caused by fires, and transfers to

the products sector (Kurz et al., 2009). Harvest transfers are directly linked to the continuation of management
practices applied within the historical period; fire emissions vary during the model run – at least for Mediterranean
countries - informed by output from climate models. No other disturbance event was considered for the period 2016
– 2100.

All main model outputs were compared with the reference scenario (RS) to estimate the average annual rate of

variation for each RCPs derived from the four climate models and the corresponding range of variation, defined as the
interval between the minimum and the maximum difference with the RS.

## 3.  Results and Discussion

Section 3.1 reports an overview of the main forest ecosystem indicators within the historical period 2000 – 2015 (RP),
as modelled by CBM, and summarizes the simulated evolution of these parameters until 2100 within the RS. Section

3.2 reports the main differences between the RS and the RCP scenarios, highlighting the additional effects of climate
change on each ecosystem driver. The combined effect of the continuation of the current management practices, as
defined within the RS, and climate change on the overall EU27+UK net $CO_2$ forest emissions, is highlighted and
discussed on Section 3.3. In Section 3.4, we compare our results with previous studies, highlight the limitations and
uncertainties of our work, and further discuss our main findings.

### 3.1. Reference Scenario: historical and long-term evolution

Net Growth (NG) is a key variable determining the evolution of all the main ecosystem indicators under different
management regimes and climatic conditions. Within the historical period 2000 – 2015 NG is equal on average to 1.7
and 1.6 t C ha$^{-1}$ yr$^{-1}$ for broadleaved and coniferous species, respectively. Since these values represent net biomass
increment before losses from disturbances, they are also directly proportional to the Net Annual Increment (NAI)

reported from NFI data and used to initialize the CBM model. This explains the lower values generally estimated for
Mediterranean and Northern European countries and the higher values estimated for Central European regions and
Ireland, at least for conifers (see Figure 1, upper panels).

NG also represents a fraction of NPP and, summing up the net litterfall, we estimated an average NPP equal to 4.4 t
C ha$^{-1}$ yr$^{-1}$ for the historical period. NPP varies from less than 2 t C ha$^{-1}$ yr$^{-1}$, estimated for the internal regions of the

Iberic peninsula, to more than 7 t C ha$^{-1}$ yr$^{-1}$, estimated for broadleaved species in central European countries (see
Figure 2, upper panels).



Under the continuation of the current management practices as detected within the historical period, CBM's output shows that NG generally decreases in most of the EU regions (Figure 1, lower panels). This is mostly due to the ongoing ageing process of existing forests, that is only partially compensated, in some regions, from management practices, which may rejuvenate the current age structure. This is, for example, the case for some Central European countries, where the NG is quite stable until the end of the century.





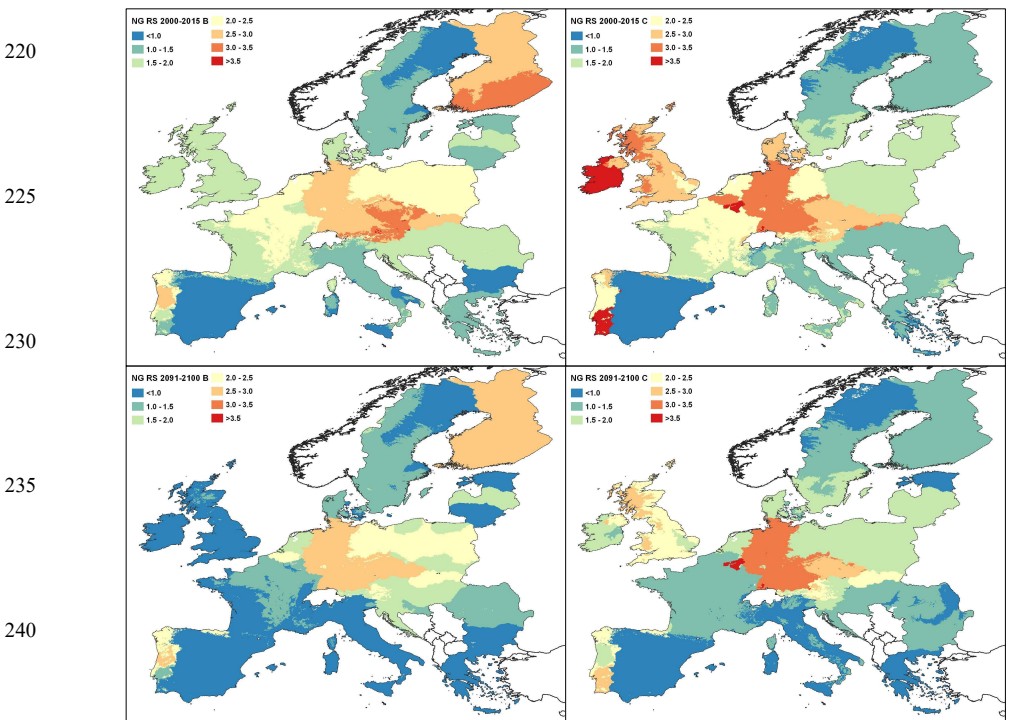

**Figure 1: geographical distribution of the average Net Growth (NG, in t C ha⁻¹ yr⁻¹) estimated by CBM within the historical period 2000 – 2015 and within the decade 2091 -2100 under the RS (i.e., excluding climate change). Broadleaved species are reported on the left side and conifers on the right side.**

Within the period 2016-2100, the share of NPP contributed by NG progressively decreases from about 38% within the historical period to 28% in 2100. However, due to the parallel increasing amount of the material loss due to the turnover rate, the average NPP increases to 4.7 t C ha⁻¹ yr⁻¹ in 2100 (+9% compared with the historical period). This is due to various, and sometimes opposite, patterns estimated on different European regions and species (see Figure 2 lower panels).



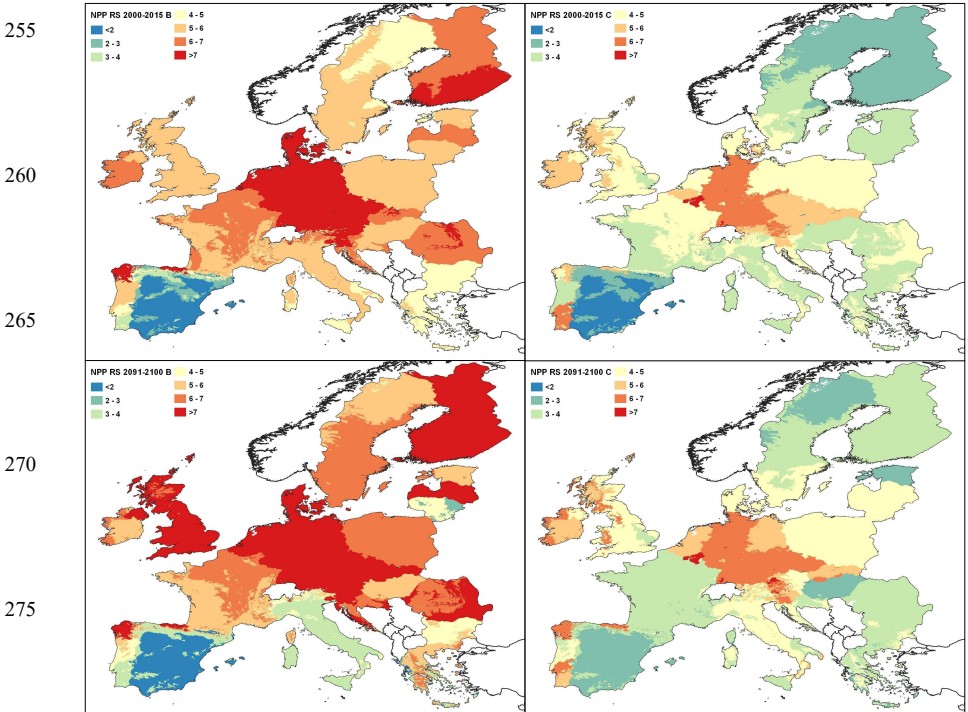

**Figure 2: geographical distribution of the average Net Primary Production (NPP, in t C ha$^{-1}$ yr$^{-1}$) estimated by CBM within the historical period 2000 – 2015 and within the decade 2091 -2100 under the RS (i.e., excluding climate change). Broadleaved species are reported on the left side and conifers on the right side.**

By subtracting from NPP all C losses due to heterotrophic respiration we estimate NEP, which represents the net change in C stocks prior to harvest or other disturbances. This is equal on average to 1.5 t C ha$^{-1}$ yr$^{-1}$ and 1.2 t C ha$^{-1}$ yr$^{-1}$, for broadleaved and conifer species, respectively, within the historical period. While NG is generally decreasing in time, under the continuation of the current management practices NEP is generally increasing in Mediterranean regions, it is quite stable in North European countries and it is partially decreasing in Central European regions and in British islands (see Fig. 8S, lower panels).

At the European level, however, because of the larger share of forest land distributed in central and north European countries, compared with the Mediterranean regions, within the period 2016 – 2100 the overall NEP decreases from about 1.3 t C ha$^{-1}$ yr$^{-1}$ within the historical period to 0.97 t C ha$^{-1}$ yr$^{-1}$ in 2100 (i.e., -28%, see Figure 3 panel A). This is due to the combined effect of a continuous reduction of the broadleaved species' NEP (-39% in 2100 compared to the historical period) and a smaller reduction (-17% in 2100 compared to the historical period, mostly after 2070) of the conifers' NEP (Figure 3, panels B-C).

We can estimate NBP by subtracting from NEP the amount of C removed by harvest and further losses due to fires. Since the absolute amount of harvest is varying on single CLUs, we cannot compare the temporal evolution of the NBP at CLU level between different scenarios, but only at EU level. Under the RS, NBP is directly affected by the





management practices applied within the historical period and their continuation, until the end of the century. Similarly at the felling rate reported from official statistics (see for example Forest Europe, 2020), the ratio between the amount

of C removed through harvest and NEP represents the intensity of the management practices carried out within a certain period. This ratio varies according to the amount of harvest reported by each country within the historical period (further corrected to account for possible inconsistencies between official statistics and other data sources), and its evolution until the end of the century (which is, in turn, determined from the evolution of the age class distribution). At European level the ratio increases from about 0.48 within the historical period (average 2000 – 2015) to about 0.77

in 2100 (see Figure 3, panel A, right axis). This is due to an increasing amount of harvest applied to broadleaved species (+25% in 2100 compared to the historical period) and, after 2065 also to conifers (+14% in 2100 compared to the historical period).

The direct consequence of this increasing harvest, combined, for both species groups, with a decreasing NEP, is a reduction by 78% of the overall NBP estimated at European level within the RS (-83% for broadleaved species and -

70% for conifers – see Figure 3).

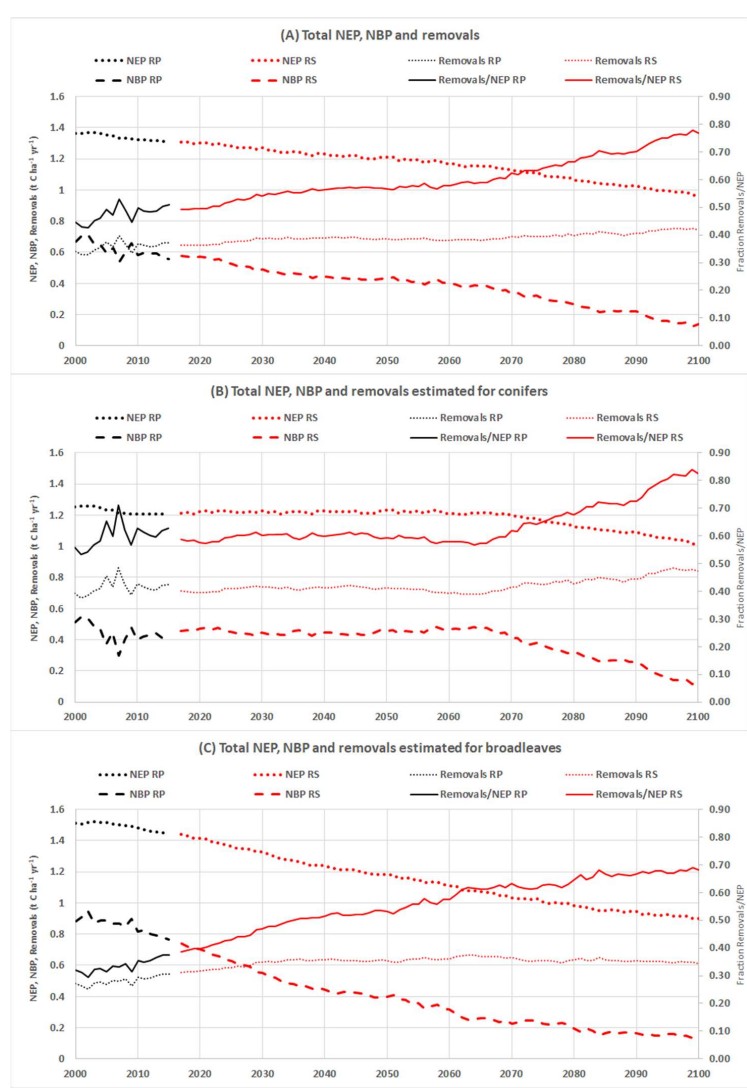

**Figure 3: Net Ecosystem Production (NEP), Net Biomass Production (NBP) and removals (all reported in t C ha⁻¹ yr⁻¹, on the left axis) estimated within the reference period 2000 – 2015 (RP) and from 2016 to 2100 under the Reference Scenario (RS) (panel A), further distinguished by conifers (panel B) and broadleaved species (panel C). The figure also reports (see the axis on the right side) the ratio between the amount of removals and the NEP, as considered within the historical period and from 2016 onwards, assuming the continuation of the current management practices detected within the period 2000 – 2015.**

### 3.2. Climate Change conditions

The additional impact of climate change on the continuation of the current management practices may partially compensate, in case of broadleaved species, or amplify, in case of conifers, the decreasing NG estimated within the



RS (Figure 4). Indeed, under both RCP scenarios, the broadleaved species' NG generally increases compared to the RS, especially in North European regions. In contrast, the conifers' NG seems to be quite stable – ranging between ±5% in comparison to the RS for most of European countries – or slightly decreasing, especially in Central-East European regions and within the first half of the century. As a consequence, at the European level, under climate

change conditions, the broadleaved species' NPP increases, compared with the RS, by 12% and 14% in 2100, under RCP 2.6 and RCP 6.0, respectively (see Fig. 9S). At the same time, the conifers' NPP decreases, by 2% in 2100, under both RCP scenarios. Due to these opposite responses, in 2100 the overall NPP increases by about 5% under RCP 2.6 and 6% and RCP 6.0, compared with the RS.



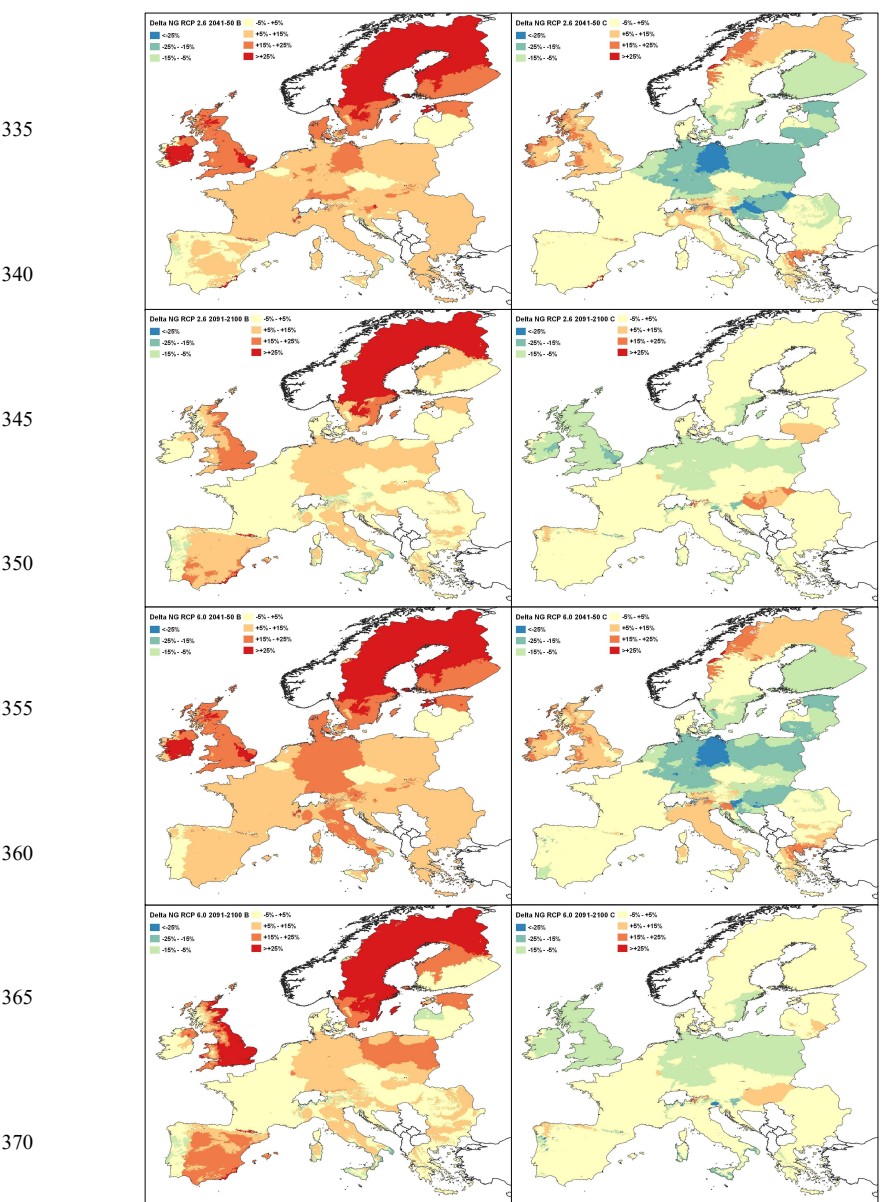









**Figure 4: relative variation of the NG due to climate change for broadleaved species (B, on the left side) and conifers (C, on the right side). The relative variation is estimated, for each country and CLU, as average percentage difference between the NG of the RS and the average NG estimated from the four climatic models within the periods 2031-2040, 2051-2060, 2071-2080 and 2091-2100. Upper four panels refer to RCP 2.6 and lower four panels to RCP 6.0.**




Such as for NG, climatic drivers also increase the broadleaved species' NPP within the entire period and under both RCP scenarios, above all within the Mediterranean and Northern European regions (see Figure 5, left panels). This
may amplify the increasing NPP highlighted under the RS (see Figure 2). Coniferous species show a different pattern, with a stable NPP within most countries, except Sweden and British islands (see Figure 5, right panels).









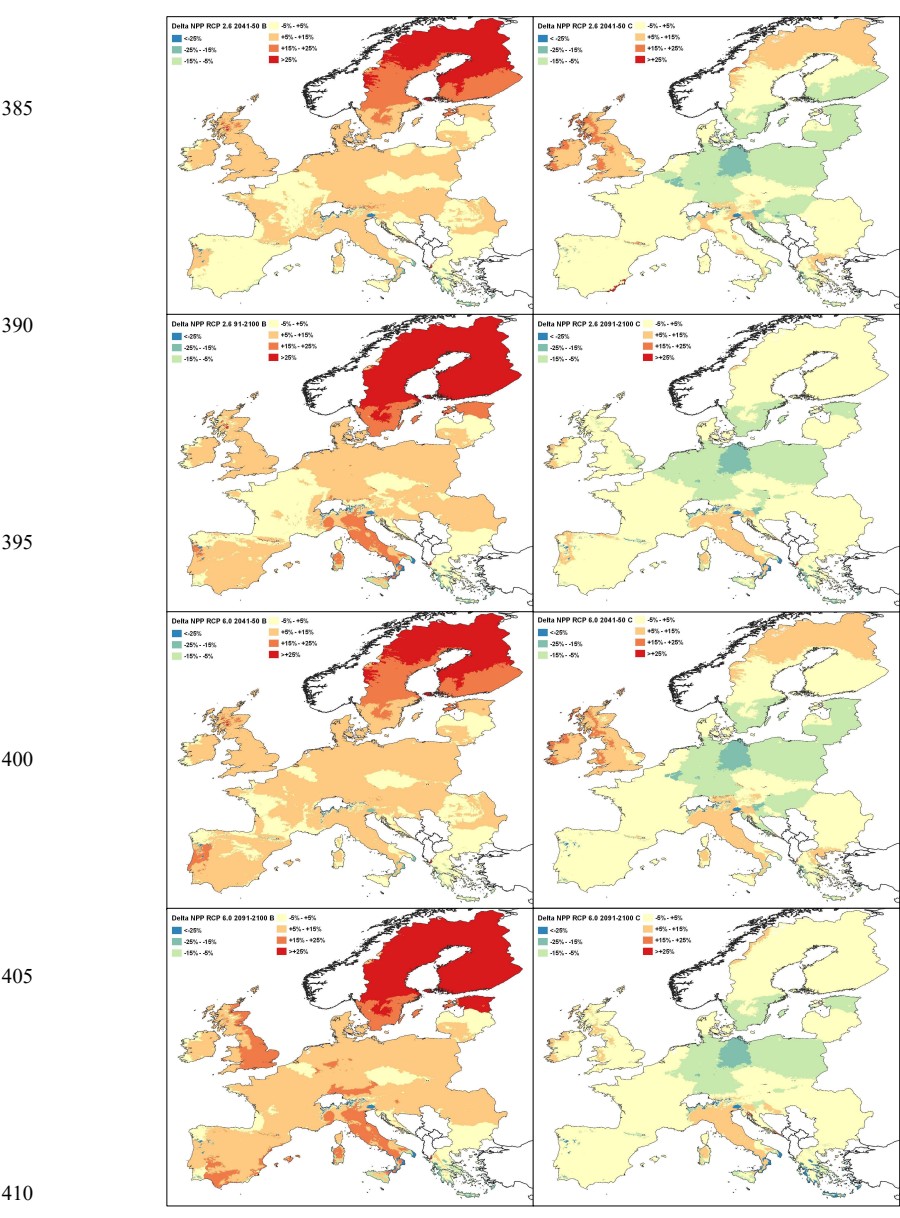

**Figure 5: relative variation of the NPP due to climate change on broadleaved species (B, on the left side) and conifers (C, on the right side). The relative variation is estimated, for each country and CLU, as average percentage difference between the NPP of the RS and the average NPP estimated from the four climatic models within the periods 2031-2040, 2051-2060, 2071-2080 and 2091-2100. Upper four panels refer to RCP 2.6 and lower four panels to RCP 6.0.**





When estimating the additional effect of climate change on heterotrophic respiration, the resulting evolution of the
        NEP becomes more complex (see Fig. 10S). For broadleaved species, we generally detected an increasing NEP until
        the period 2071- 2080, with the exception of some specific CLUs. This means that the combined effects of climate
        changes on net growth and heterotrophic respiration may compensate the decreasing NEP. By the end of the century,
        however, especially under RCP 2.6 this trend could be attenuated or even be reversed, at least within some East
European countries, and in some other regions. Moreover, when considering the maximum and minimum values
        derived from single climatic models, our analysis highlights strong interannual variations of the NEP, from +60% to
        -40% for broadleaved species (Figure 6). This suggests that the effect of climate change may overcome, in single
        years, the evolution of NEP due to biological processes and forest management practices. For conifers, NEP is
        generally quite stable or decreasing within all the European regions and under both RCP scenarios, except the Italian
peninsula for the entire period and British islands until 2050 (see Fig. 10S). This means that climate change may
        amplify the loss of C stored within the coniferous forests - or potentially available for harvesting – reducing, by the
        end of the century, the average NEP by about 7% and 8%, under RCP 2.6 and RCP 6.0, respectively (Figure 6). The
        overall effect at the European level is a compensation between different regions with opposite trends, at least until
        2090, when, especially under RCP 2.6, we estimated a percentage reduction of the average NEP, equal to -10% in
2100. In all cases, however, we highlighted that interannual variation of the average NEP estimated under the RS due
        to the effects of climate change is considerably larger than the effects detected on NPP (see Fig. 9S and Figure 6).
        This is due to the combined effects of climate variables on NG and heterotrophic respiration.



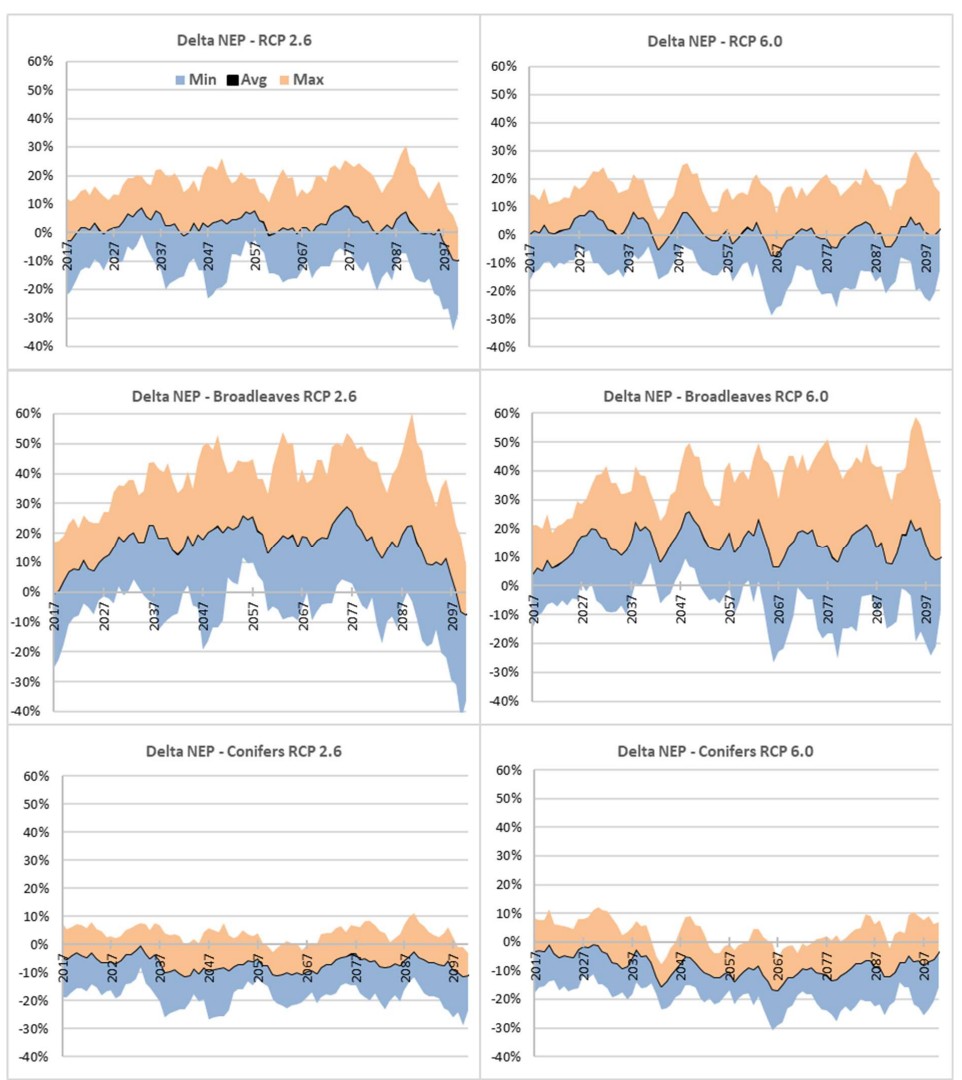

**Figure 6: average annual rate of variation of the Net Ecosystem Production (NEP), compared to the RS, derived from the four climate models, at EU level (upper panels), for broadleaved species and for conifers, under different RCP scenarios. Minimum (Min) and maximum (Max) percentage values correspond to the interval between the minimum and maximum difference with the RS for each year. All values are reported as 5-year moving averages.**

Considering the additional effect of harvest (which is not varying between reference and climate scenarios) and wildfires, until 2090 the resulting NBP estimated at European level is slightly increasing (on average +11% between 2016 and 2090 under RCP 2.6 and +6% between 2016 and 2090 under RCP 6.0 - see Figure 7). Within the last decade of the century, however, under RCP 2.6 we estimated a further marked reduction of NBP equal to about -14% between 2091 and 2100, while under RCP 6.0 we estimated the opposite pattern (+18% between 2091 and 2100). This is the





result of the opposite effects of climatic impact on the NBP dynamics of broadleaved species and conifers. Indeed, despite strong interannual variations, climatic drivers generally increase NBP estimated for broadleaved species – at least until 2090 - and slightly decrease NBP estimated for conifers.

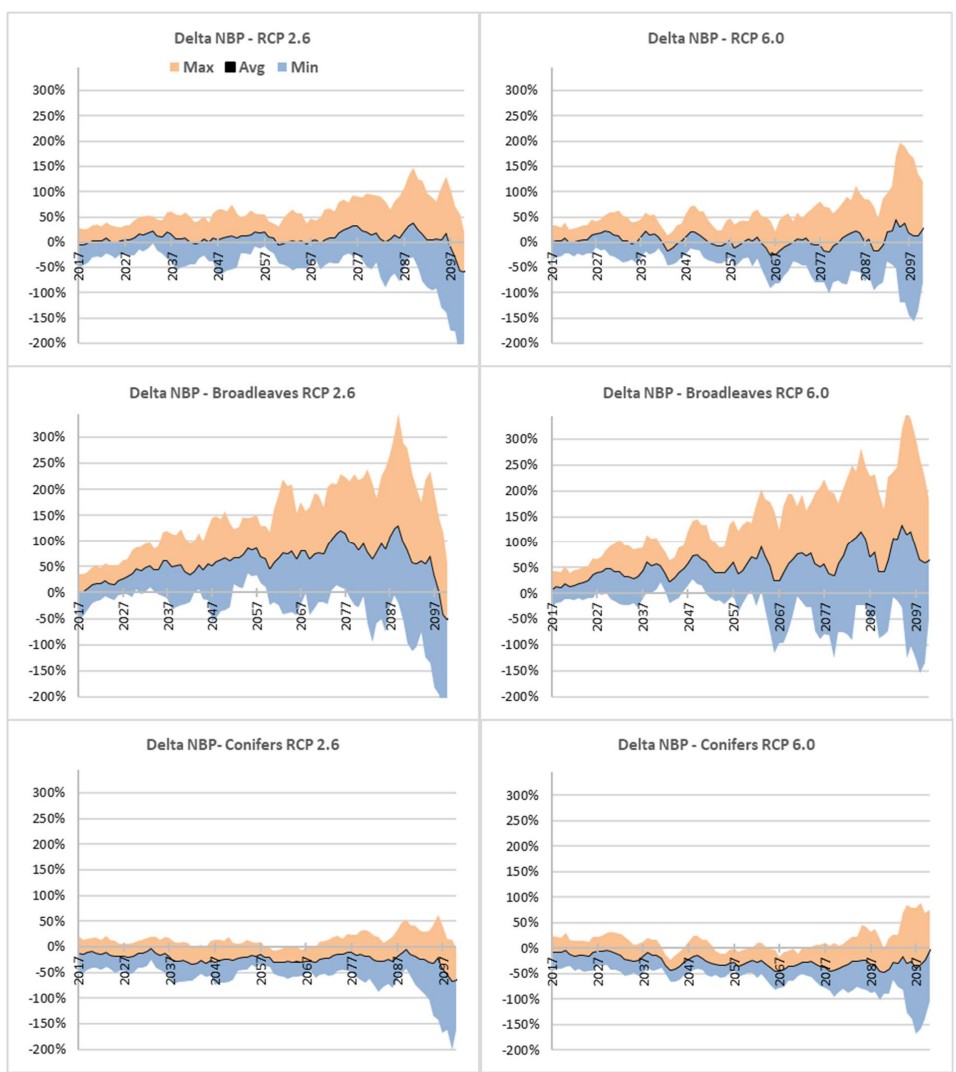

**Figure 7: average annual rate of variation of the Net Biomass Production (NBP), compared to the RS, as derived from the four climate models, at EU level (upper panels), for broadleaves and for conifers, under different RCP scenarios. Minimum (Min) and maximum (Max) percentage values correspond to the interval between the minimum and maximum difference with the RS for each year. All values are reported as 5-year moving averages.**



### 3.3. Total Net CO2 emissions

The long-term dynamic of the $CO_2$ emissions estimated within the RS is mostly driven by the changes in the age-class

distribution and by the specific management practices applied within the period 2016-2100. These practices, directly determine the net CO2 emissions, because NBP is calculated from the difference between NEP and removals (plus other losses due to natural disturbances). Forest management also affects the annual growth rate, modifying both the age-class distribution - through clear cuts or single tree selection systems - and the overall density of the forest stands – through thinnings. Assuming the continuation of the management practices applied between 2000 and 2015, we

estimated a reduction in the forest sink, from about -353 $CO_{2eq}$ $yr^{-1}$ within the historical period (average 2000 – 2015) to -79 Mt $CO_{2eq}$ $yr^{-1}$ in 2100 (i.e., -78%, see Figure 8). When considering the additional effect of climate change, since coniferous species - where we estimated a decreasing NBP - cover about 60% of the total forest area, the overall total net CO2 emissions decreases to -34 Mt $CO_{2eq}$ $yr^{-1}$ estimated within RCP 2.6 in 2100 (i.e., -57% compared to the RS). The average trend based on RCP 6.0 is similar to the one estimated under the RCP 2.6 even if in both RCP scenarios,

interannual variations largely exceed the difference between the RS and the average derived from the four climate models. This is due to the large uncertainty of climate projections. For clarity, Figure 8 highlights only the range between the minimum and maximum values of the ensemble of climate models estimated under RCP 2.6. Similar results for RCP 6.0, with a larger magnitude of variation of the net CO2 emission estimated under climate change conditions, are reported on Fig. 11S.


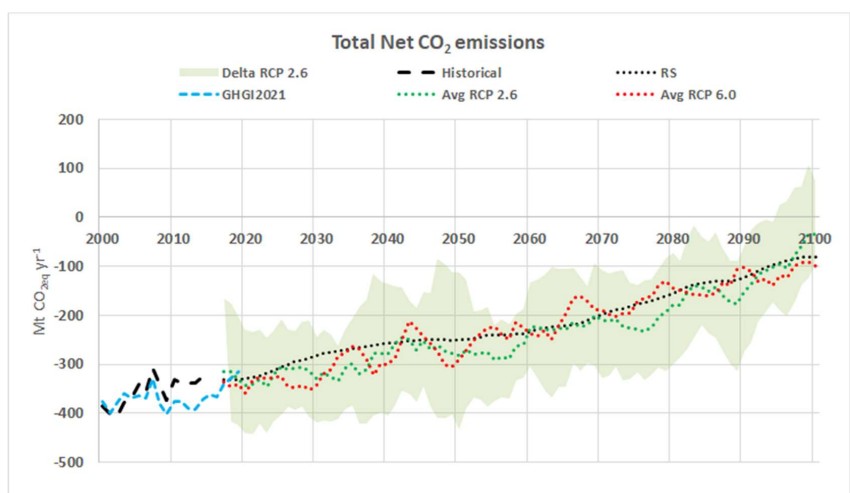

**Figure 8: total net CO2 emissions (reported as CO2eq. yr-1, with negative values conventionally highlighting CO2 removals from the atmosphere) estimated within the historical period, under the reference scenario (RS), and under RCP 2.6 and**

**RCP 6.0 (reported as the average values estimated from different climate model within each RCP scenario). The figure also reports the net emissions reported from EU27 + UK Member States, according to the Greenhouse Gas Inventory 2021 (GHGI 2021, referred to the category Forest Land Remaining Forest Land, as reported from UNFCCC CRF Tables, 2021), and the range between the minimum and maximum values estimated under RCP 2.6. All values derived from the present study are reported as 5-yr moving averages.**





**3.4. Comparison with other studies, limitations, and uncertainty of the present study**

NPP is a key variable for understanding the forest carbon cycle and for assessing the potential timber supply, as affected by climate change. There are different data sources and methods to assess NPP: process-based models, such as the ones used within the present study, remote-sensed approaches, such as MODIS NPP (Running et al., 2004), inventory-based models, like CBM (Kurz et al. 2009) or EFISCEN (Schelhaas et al., 2007), and indirect estimates
based on field measurements provided from NFI data. Of course, each approach has specific pros and cons, and different studies have compared various estimates at European and country level. Neumann et al. (2016) developed a regional MODIS-NPP dataset for the European forests, named MODIS EURO, combining remotely sensed satellite - driven data, with terrestrial NFI data and tree carbon estimations. Based on this assessment, Neumann et al. (2016) estimated an average NPP equal, at European level, to about 5.8 t C ha$^{-1}$ yr$^{-1}$and 5.4 t C ha$^{-1}$ yr$^{-1}$, according to MODIS
EURO and NFI data, respectively. Similar results are reported, for the period 2000 – 2012, from Hasenauer et al. (2017), combining MODIS EURO data with the field measurements provided from 13 NFIs. These values are generally higher than previous estimates provided by Tupek et al. (2010), who compared the average forest NPP estimated from EFISCEN for 2005 with three different process-based models. These authors report an average NPP for 26 European countries (mostly overlapping with the present study) equal to about 5.1 t C ha$^{-1}$ yr$^{-1}$ (with a SD=1.8),
4.9 t C ha$^{-1}$ yr$^{-1}$ (with a SD=1.2), 5.5 t C ha$^{-1}$ yr$^{-1}$ (with a SD=1.6) and 4.2 t C ha$^{-1}$ yr$^{-1}$ (with a SD=0.9), based on EFISCEN, BIOME-BGC, ORCHIDEE and JULES models, respectively.

The average NPP estimated by CBM in our study, equal to 4.4 t C ha$^{-1}$ yr$^{-1}$ within the period 2000 – 2015, lies within the range of values reported by Tupek et al. (2010), even if it is generally lower. The differences between various approaches are further amplified when comparing these estimates at a country level (see Fig. 12S). This is due to
various reasons. First, some models, such as MODIS EURO, also cover non-forest lands such as crops, shrubs or grasslands (Neumann et al. 2016). Other models, such as EFISCEN or NFI-approaches, may have been mostly calibrated against data collected within the Forest Area Available for Wood Supply (FAWS), where increment and NPP values may differ from unmanaged forest lands. Within the present study, we also considered about 12 M ha of unmanaged forest lands, mostly located within Northern European countries and the Iberic peninsula and generally
having a lower NPP. This can also explain the differences with a previous study, always based on the application of the CBM model at European level (Pilli et al., 2017), reporting an average NPP equal to about 5.1 ±1.4 t C ha$^{-1}$ yr$^{-1}$. This last value, however, was not estimated from a spatial distribution of the NPP between different CLUS, such as on the present study, but from the average NPP values estimated at country level. If considered at the same way, the average NPP estimated within the present study, equal to 5.2 t C ha$^{-1}$ yr$^{-1}$, is well in line with the previous estimates
based on the CBM model. Natural disturbances, such as windstorms and wildfires are not directly considered in EFISCEN, except if directly affecting removals through salvage logging, and they may also have not been considered by process-based models or by NFI data, which refer to specific time intervals, generally within the period 2000 – 2010. In contrast, all major disturbance events affecting the European forests within the historical period were included in our model runs. Refining the representation of fires and other natural disturbances, may considerably improve the
estimates reported from earth system models, which are probably overestimating the forest biomass C accumulation, at least within boreal ecosystems (Wang et al., 2021).



The results obtained from process-based models may also be partially biased because of the spatial distribution of FLUXNET forest sites, mostly concentrated in western, northern, and middle European countries (Tupek et al., 2010). However, CBM results - like those from EFISCEN - are strongly affected by the quality of input data, including both

NFI measurements of volume and increment, and harvest statistics (see Pilli et al., 2016). As noted by various authors, information on harvest reported from official statistics are, in many cases, largely biased (see for example Camia et al., 2020). For this reason, in our study official statistics reported from FAOSTAT (http://www.fao.org/faostat/en/#data/FO ) were further compared, and eventually corrected, according to other data sources (Pilli et al., 2015). Nevertheless, some recent additional information provided by countries on harvest,

increment and forest management, was not included in our assessment, but they could further improve our analysis, above all for the historical period 2000 – 2015 and the definition of the RS. (Korosuo et al., 2021). Other factors, such as the quantification of litter fall and fine root turnover rates, may explain the differences between our results and other estimates, based for example on NFI data. Neumann et al. (2016) derived the total litterfall from a meta-analysis based on 471 Eurasian stands, as reported from Liu et al. (2004). For boreal and temperate forests (further

distinguished between continental, mountain and oceanic), Liu et al. (2004) report an average total aboveground litterfall ranging from a minimum of $1.9\pm0.8$ t C ha$^{-1}$ yr$^{-1}$ and a maximum of $3.5\pm1.1$ t C ha$^{-1}$ yr$^{-1}$. For the historical period, we estimated an average litterfall equal to about 2.6 t C ha$^{-1}$ yr$^{-1}$, which lies within the range reported from these authors, but it also includes belowground biomass turnover. Despite the differences between the absolute NPP values reported by different authors, we notice that the spatial distribution reported for the historical period by our

study (Figure 2), is mostly in line with the results provided from Hasenauer et al. (2017, *cfr* Fig. 2 in that study) and Neumann et al. (2016, *cfr* Figure 2 in that study) even if these studies do not distinguish between broadleaved species and conifers. Of course, in our case, since the CBM model is not spatially explicit, the spatial resolution of our estimates is limited to the spatial scale attributed to each CLU, as considered at country level within our study and it was not further interpolated with a forest map. Integrating these results with other maps reporting forest composition

and biomass distribution may certainly improve our estimates (Avitabile et al., 2020).

When analyzing the long-term evolution of NPP under the RS, while the average NG decreases, from about 1.7 t C ha$^{-1}$ yr$^{-1}$ within the historical period to 1.4 t C ha$^{-1}$ yr$^{-1}$ in 2100, the absolute amount of litterfall increases from 2.6 t C ha$^{-1}$ yr$^{-1}$ to 3.3 t C ha$^{-1}$ yr$^{-1}$ in 2100. For this reason, the share of NPP contributed by material loss progressively increases, from about 62% within the historical period to 72% in 2100. This is mostly due to the ongoing ageing

process, which is increasing the biomass standing stock, but it is progressively decreasing the percentage annual biomass increment (see Fig. 13S). Indeed, when forest stands are getting older, a larger proportion of NPP is allocated to the replacement of the material lost to turnover (Köhler et al., 2008). Because of these opposite but interdependent trends, the final NPP increases to 4.7 t C ha$^{-1}$ yr$^{-1}$ in 2100. By subtracting the losses due to the heterotrophic respiration, the total amount of C potentially available for storage or for harvesting in 2100, i.e. NEP, decreases within most of

the European regions (see Fig. 8S). Similarly to NPP, the average NEP estimated by CBM within the historical period, equal to 1.3 t C ha$^{-1}$ yr$^{-1}$, is generally lower than the values estimated from eddy covariance measurements ($2\pm2.6$ t C ha$^{-1}$ yr$^{-1}$) or from NFI data ($1.6\pm0.2$ t C ha$^{-1}$ yr$^{-1}$) (see Luyssaert et al. 2009). Our estimates, however, are within the range of values reported from Zaehle et al. (2006) through the application of a modified LPJ approach (NEP= $1.3\pm0.4$



t C ha$^{-1}$ yr$^{-1}$). Interestingly, while coniferous' NEP is quite stable until 2070, broadleaved species show a continuously
decreasing NEP within the entire period (Figure 3). Taking into account the ongoing evolution of the living biomass
C stock and of the corresponding percentage increment (as reported on Fig. 13S), this may suggest that, due to
management practices, the age-class distribution of broadleaved species stands ages faster than that of conifer stands.
Indeed, assuming the continuation of the current management practices until 2100, part of the broadleaf's forest area
will not be rejuvenated, and the current age structure is projected to get considerably older than coniferous' age
structure and data derived from NFI measurements. For this reason, in our study, the growth functions derived from
NFI data, were also preliminarily updated, to mimic the long-term evolution of NPP in aging forests, according to the
data reported from Tang et al. (2014) - see section B in Supplement Materials and Fig. 2S. The potential old-growth
forest C sink is still debated within the literature (Gundersen et al., 2021; Luyssaert et al., 2021), and our estimates
confirm that with increasing age, these forests show a decreasing net biomass accumulation rate.
Considering further losses due to harvest and natural disturbances, we estimated an average NBP equal to 0.60 t C ha$^{-1}$ yr$^{-1}$ within the historical period. Apart from comparing this value with similar estimates reported from previous
studies (see Luyssaert et al., 2009), we can also calculate the corresponding total net CO2 emissions of EU27+UK.
As highlighted on Figure 8, our estimates are well in line with the net CO2 emissions reported from EU27+UK
countries within their Greenhouse Gas Inventories (GHGIs, as reported from UNFCCC CRF Tables, 2021), with an
average difference equal to about -6%, mostly concentrated within the period 2008-2015. Part of this difference is due
to slightly different assumptions about the forest area in our study and those reported in countries' GHGIs, and part
of that is due to different assumptions about the amount of harvest - including the share of salvage logging - and about
the impact of natural disturbances, in particular after 2010. Despite that, however, it is important to notice that the
absolute net emissions reported in the RS after 2015, are in line with the recent values reported from GHGIs for the
period 2017 – 2019, even if they could be based on a slightly different amount of harvest.
Comparing our estimates with other studies considering the dynamics of the European forests to the end of the century,
under different management regimes and climatic conditions, is more challenging because, at least to our knowledge,
there are not many studies assessing both these aspects together and within such a long time-horizon. Some studies
assessed different forest management regimes until 2030 (Rüter et al., 2016; Jonsson et al., 2021) or 2050 (Nabuurs
et al., 2017), but they did not consider climate change. Reyer et al. (2012), analyzed the forest productivity change,
under different climatic conditions, for four tree species and ten environmental zones in Europe, using the process-
based model 4C calibrated against the data provided from three different climate models and two different assumptions
about CO2 effects on productivity. According to their results, Northern Europe productivity – dominated by Scots
pine and Norway spruce - will generally increase, while Southern Europe productivity will mostly decrease. These
results are not fully in line with our estimates and other studies (i.e., Sperlich et al., 2020). Indeed, as reported in
Figure 5, for broadleaved species, we predicted an increasing NPP both within the Mediterranean regions and Northern
European countries, and for conifers a quite stable NPP, within Western European countries, or decreasing NPP, in
Central European countries. Interestingly, the tipping point when initial gains in NPP turned into losses, that we
detected at European level in coniferous species around 2030, was also noticed within a recent study, analyzing forest
productivity in Germany, under climate change conditions (Sperlich et al., 2020).



Other nationwide studies, based on direct field measurements, are well in line with our results. For example, Bosela et al. (2021) highlight the negative impact of climate warming and other environmental factors, across to a biogeographical gradient, on Norway spruce productivity in Central European regions. Various reasons may explain the differences between these studies. Reyer et al. (2012) focused on the physiological response to global change, and they did not consider Mediterranean tree species, but only boreal and temperate species (European beech and oak) that occur in the Mediterranean regions. Most importantly, our results account for the combined effect of forest management and climate change, while previous studies mostly focused either on different management strategies or climate change conditions. Combining both these aspects within a hybrid modeling framework constitutes, in our opinion, the main added value of our work.

Simulating the continuation of the current management practices from the historical period, and excluding additional effects due to climate change, we estimated a CO2 net C sink decreasing from -331 Mt $CO_{2eq}$ yr$^{-1}$ in 2016 to -79 Mt $CO_{2eq}$ yr$^{-1}$ in 2100. Part of this reduction is certainly due to the amount of C removed with management in the RS, increasing from about 100 Mt C yr$^{-1}$ within the historical period, to 118 Mt C yr$^{-1}$ in 2100.

A fraction of these removals, which amount to about 9200 million tons of C (as cumulated value, removed under the RS between 2016 and 2100), will be temporarily stored within the Harvested Wood Products pool, which is not considered within the present analysis but may only partially compensate the reduction of the forest C sink directly provided by forest ecosystems (Jonsson et al., 2021).

The ongoing ageing process of the European forests, however, plays a key role. Indeed, as reported on Figure 3, while the NBP of coniferous species shows a stable trend until 2065 then decreasing according to the increasing amount of removals, the NBP of broadleaved species continuously decreases throughout the entire period, despite the stable amount of removals applied within the RS (equal to about 0.6 t C ha$^{-1}$ yr$^{-1}$ from 2030). This suggests that, at least for broadleaved species, reversing this process to maintain or enhance the current forest mitigation potential, will be quite challenging. This could also be the result of the different management regimes for this group of species within the last decades, including for example the abandonment of large coppice areas within many Mediterranean regions (Müllerová et al., 2015).

Our findings, are clearly affected by our methodological assumptions, including the frequency and intensity of specific management practices applied at the country level, the reliability of the age-class distribution as considered by our model and of the growth functions applied within the model run, as derived from NFI data. In particular our assumptions about the long-term dynamics of uneven-aged forests- mostly distributed within the Mediterranean countries – and about the real impact of stand-replacing management practices - which can rejuvenate the current age structure - could have reduced the final NG, and as a consequence also the NBP, estimated within such a long-term model run. Despite these uncertainties, however, our results are substantially in line with the main findings proposed by other studies carried out both at national (e.g., Jandl et al., 2018) and at the European level. Assuming the continuation of the management practices and harvest intensity detected within the period 2013-2017, Welle et al. (2020) estimated a biomass C sink equal to about -245 Mt $CO_{2e}$ yr$^{-1}$, which is consistent with our results (Böttcher and Frelih-Larsen, 2021). Valade et al. (2017) assessed the optimal forest management strategies for mitigating climate change using a conceptual empirical model of sequestration efficiency and concluded that, in a long-term, the EU





forest sector (including HWP and material substitution benefits) remains a net C sink in 99% of the simulations, but in 25% of the simulations the forests themselves become a source and only in 25% of the simulations the sink
efficiency was found to be enhanced. Overall, all these studies, including the main finding our study, suggest the urgency to develop management strategies to partially reverse the declining C sink that is expected under the continuation of current management practices.

Climate change could amplify or mitigate this ongoing trend. At the European level the impact of climatic drivers could be negligible on the long-term dynamics of NPP, at least if compared with the ongoing changes of the age-class
distribution (see Fig. 9S). The impact of climate change, however, would certainly be higher on NEP (see Figure 6) and on NBP (see Figure 7), because the cumulative effects of different climatic drivers on net growth, heterotrophic respiration and on natural disturbances - even if these were limited in our study to changes in the fire frequency in Mediterranean countries. As a result, the total net $CO_2$ emissions estimated at European level – and even more if considered at country level - may largely vary because of the interannual variations due to climatic variability and
stochastic natural disturbances (see Figure 8). Different, and sometimes opposite climatic impacts on broadleaved species and conifers, and in different regions, suggest that, in some cases, substituting the current species, for example increasing the share of broadleaved species in Central and Mediterranean regions (see also Bosela et al., 2021), could be part of specific forest management strategies developed at regional and local levels (see Nabuurs et al., 2018). Apart from modifying the current forest composition – which was assumed as constant within our study (see Morin
et al., 2018) -, other options to rejuvenate the current age structure may include, for conifers, a gradual shift towards a continuous cover forestry system based on an uneven-aged structure (Valkonen et al., 2020), and for broadleaved species a gradual recovery of past management practices, dismissed, at least in some regions, after the Second Word War (Müllerová et al., 2015). This could partly compensate the continuously declining percentage C stock change estimated for broadleaved species (see Fig. 13S). On the other hand, the percentage C stock change of coniferous
species is quite stable until 2070, despite the higher amount of removals (see Figure 5, panels B and C, and Fig. 13S). Both these solutions could also provide other additional ecosystem benefits, but they need to be further assessed, as part of a broader forest strategy, which also includes protecting primary and old-growth forests (as stated within the EU Biodiversity Strategy), new afforestation activities (such as the so called "3 billion Trees" initiative promoted within the new EU Forest Strategy), restoring existing forests, and reducing the impact of natural disturbances (EC,
2020a, 2020b, 2021). In particular, the expected increasing impacts of windstorms, insect outbreaks and wildfires in Central and Northern European countries, were not considered in our study, but these will further reduce the future forest C sink (Forzieri et al., 2021; Senf and Seidl, 2021).

Our results may also be biased by the gap between the growth functions applied by CBM, based on input data derived from NFI field measurements – in particular NAI - and the current growth of the forest species. In some cases, NFI
data are quite outdated, since they may be based on NFI concluded between 2005 and 2010 (or, in few cases, even before 2005). As a consequence, the growth functions derived from these increment data do not properly consider the most recent direct effects of climate change on the current growth rate. As highlighted by some studies (see for example Bosela et al., 2021), these effects are already quite evident in some European regions, and they should be properly represented in inventory-based models, such as CBM, that are mostly based on NFI data.



### 4. Conclusions


Our study successfully combines a stand-level, inventory-based model, particularly suited for simulating various forest management strategies and disturbance regimes, with the output provided by a DGVM, driven by four process-based climate models. This meta-modelling approach highlights that, under the continuation of the current management practices, the EU27+UK forest C sink will be reduced by about 77% by the end of the century. The additional effect
of climate change may either amplify or mitigate this trend at the local level, resulting in strong interannual variations, which may double, or half, the EU-wide forest C sink. The impact of climatic drivers, generally lower on NPP, and gradually larger on NEP and NBP - because of a cumulative effect on various physiological processes and disturbances - may vary, according to the species composition and the geographical impact of climate change. Both RCP scenarios yield a similar pattern, in particular in the first half of the century. In some cases, the combined effects of these factors
on net growth and on heterotrophic respiration may compensate the decreasing NEP due to the aging process that results from the continuation of the current management practices.

To become climate neutral by 2050, the EU27 net C sink from forest land should increase to -450 Mt $CO_{2e}$ $yr^{-1}$ by 2050 (EC, 2020a), but, according to our estimates, under the continuation of the forest management regime applied within the period 2000 – 2015, this sink (including UK) would decrease to about -250 Mt $CO_{2e}$ $yr^{-1}$ in 2050. These
results are consistent with the main findings from other studies (Valade et al., 2017; Welle et al., 2020). By assuming additional mitigation initiatives, such as different management regimes and a further expansion of the forest area, other studies report a potential increasing forest C sink ranging between -150 and -400 Mt $CO_2$ $yr^{-1}$ in 2050 (Nabuurs et al., 2017, EC, 2020a). These figures do not account for the possible increasing impact of climate change and natural disturbances, but they include the additional mitigation potential provided from carbon storage in harvested wood
products and material and energy substitution. While both these elements were not considered in our study, it is unlikely that they would compensate for the reduction of the C sink directly provided by forest ecosystems (Jonsson et al., 2021; Köhl et al., 2021).

The main driver of the long-term dynamics of the forest C sink seems to be the ongoing ageing process of the European forests, mostly determined by historical management (McGrath et al., 2015) and current silvicultural practices (e.g.,
harvest) - and partly by our specific methodological assumptions (i.e., on forest management and uneven-aged forests). Climate change, however, apart from contributing to strong interannual variations, may further reduce the EU forest net C sink or mitigate this trend. Due to the uncertainty about the future evolution of environmental variables and the relative impact of these variables on forest growth and mortality, in 2050 the EU 27+UK forest net C sink may range from about -100 Mt $CO_{2e}$ $yr^{-1}$ under RCP 2.6 and 6.0 to about -400 Mt $CO_{2e}$ $yr^{-1}$ and –300 Mt $CO_{2e}$ $yr^{-1}$, under RCP
2.6 and 6.0, respectively. This means that, reversing this process, to maintain or enhance the current forest mitigation potential, will be quite challenging and urgently requires alternative management strategies (Yousefpour et al., 2017) and new modelling tools, that merge traditional scientific objectives – generally linked to a climate change perspective – with practical applications to forest management and planning activities (Shifley et al., 2017).

In this sense, despite the uncertainty and some methodological limitations of this study – excluding, for example, the
additional impact of windstorms and insect outbreaks or possible effects on trees' species composition - our results may constitute a first benchmark to set up specific management strategies, defined at European level, and further



downscaled at regional and local level. This is in line with the new EU forest strategy, where climate change mitigation and adaptation should be part of a broader roadmap, including biodiversity, conservation and a sustainable use of forest resources (EC, 2021a). Failure to achieve the planned forest sinks by 2050 will make achieving net zero goals

even more difficult.

**Code and data availability**. The EU Archive Index Database used by CBM model, such as detailed information on the spatial distribution of climatic units is available at https://data.jrc.ec.europa.eu/dataset/jrc-cbm-eu-aidb. Other data input and simulation outputs and used for this study are available upon request to the corresponding author.

**Author contributions**. RP and RA designed the methodology, in collaboration with AC, WK and GG; RA carried
out the data analysis on the land-climate models and described the corresponding methodological approach; RP carried out the analysis on the forest growth model and wrote the paper, in collaboration with RA, AC, WK and GG who helped in the interpretation and discussion of the results. All authors read and approved the final paper.

**Competing interests.** The authors declare that they have no conflict of interest.

**Disclaimer.** The views expressed are purely those of the authors and may not in any circumstances be regarded as
stating an official position of the European Commission or Natural Resources Canada.

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
