# Peer review of "The European forest Carbon budget under future climate conditions and current management practices"

_Biogeosciences, 2022_

## Referee Comment (RC2)

Review of manuscript submitted to Biogeosciences on "The European forest Carbon budget under future climate conditions and current management practices"

**General comments**

My review of the manuscript BG-2022-35 by Pilli et al. finds that the paper is of high scientific significance. It can be considered an important contribution to scientific progress in the field of forestry research and environmental policy as it provides a concept for combining different types of models for addressing urgent policy questions. The paper is of high scientific quality and to my knowledge includes the most recent and relevant literature on the topic. There are limitations to the approach and the study leaves open questions, e.g. on the interaction of forest management and natural disturbances, the effect of other disturbances beyond fire, the effect of management changes. However, the authors are not tempted to overload the study but focus on practicability of the approach. This has of course also limitations for the interpretation of the results for policy. And here is my only criticism: in the paper the authors draw policy conclusions like the study "may constitute a first benchmark to set up specific management strategies". Due to the limitations of the modelling approach and the rather crude assumptions on the reference scenario conclusions on needed responses to revert the declining trend should not be drawn. As the authors mention in their response to Anonymous Referee #1 (https://doi.org/10.5194/bg-2022-35-AC1), "the continuation of forest management (BAU) was chosen just to test our method, but this is not a policy scenario". In that sense the manuscript should more carefully draw conclusions on how to respond to the scenario results. Instead, the authors could provide requirements for making the results more policy relevant, e.g. by a more policy-oriented scenario design, sensitivity analyses regrading forest management options etc.

Overall, the manuscript is well structured, the language clear and methodology and assumptions well-presented.

**Specific comments**

- Lines 205 ff: This explains… This is not clear to me: please more explicitly explain the geographical differentiation in the RC 2000-2015
- Line 213 ff: This is… You attribute the reduction of NG "partially" to "ageing". I think this needs to be verified and better supported by data. Also the "rejuvenation" might reduce NG if it moves the biomass stock of a stand below the maximum increment. That there is instead a saturation effect can be observed from Figure 13S. I suggest adding a sentence on these dynamics and refer to this figure at this point as the biomass stock/increment relations and dynamics cannot be observed from Fig 1.
- Lines 245 ff/Fig 1: The labels of what the panels show are very small. I suggest adding "upper panel right:…" etc. to the figure caption for better readability.
- Lines 280 ff/Fig 2: The labels of what the panels show are very small. I suggest adding "upper panel right:…" etc. to the figure caption for better readability.
- Lines 289-291: At the European… The sentence is hard to understand. I suggest splitting it into two for more comprehensiveness.
- Section 3.4. "Comparison with other studies, limitations, and uncertainty of the present study" is very long. It addresses different aspects that should rather be separated for more readability and clearer structure. Suggestions for additional sections are: Comparison of the reference period with other data sources, Assessing impacts of climate change, Assessing underlying trends of growth dynamics, Limitations of the model, …

- Lines 692-695: Due to the… The expected ranges for RCP 2.6 and 6.0 for 2050 are not clear from this sentence. Please revise it, e.g., by using the formulation in the abstract where it is much clearer.

**Technical corrections needed**

- Line 65: so-called
- Line 79: combining
- Caption Figure 1S: LP-GUESSJ should be LPJ-GUESS
- Line 117: delete by year
- Line 155-157: Sentence starting "This is…" unclear, please revise.
- Line 160: Suggest naming the output variables after "output" again (Growth Multipliers and area burnt)
- Line 568: in Figure 8
- 570-573: language, consider splitting sentence into two

---

## Author Response (AR1)

Editor's comments:
Thanks for your submission to Biogeosciences. This manuscript was read by two referees, who both provide thoughtful and in-depth comments. Referee 1 is quite positive overall, but suggests presenting results on stocks as well as fluxes in the main text, and noting that Rh can be decoupled from NPP.

**Many thanks for your positive comments and for your constructive suggestions. To address the comments provided from Rev #1, we moved from supplementary materials to the main text a figure (now reported as Figure 4) reporting the long term evolution of the living biomass C stock determined under the Reference Scenario (this also address a specific comment provided from Rev #2). The trend and the absolute level of C stock reported on this figure for the RS do not diverge from the C stock determined under climate change scenarios.**
**Moreover, even if we do not consider strictly needed for our analysis reporting the evolution of Rh (please see our comments below), we added, as supplementary material, a specific figure reporting the evolution of Rh both under the reference scenario and climate change scenarios (see Fig. 8S). Both these figures were further discussed within the main text (please see replies to Rev #1 and Rev #2 below)**

Referee 2 has a similar assessment, but also points out the inconsistency between limitations of the modeling approach, your declaration that "this is not a policy scenario", and some of the language in the text. I think this point is well founded. Both referees are quite positive overall, and both also provide many detailed comments and suggestions.

**We fully understand your point and agree with the comment provided from Rev #2, therefore we further highlighted both within the abstract and in the main text (see L 21-22, 136-137, 228-229, 785) that, our reference scenario, such as both the climatic scenarios linked to the previous one, aim to present a modelling framework and not a policy scenario.**

I have read the manuscript and broadly agree with the reviewers. This is a well-written and interesting analysis overall, but it does need substantial revisions in many areas to address the issues raised by R1 and R2; I have reviewed your responses and am convinced there's a solid path forward in this regard. My one additional suggestion is that section 4 (essentially the discussion) is very long relative to the rest of the ms; please look for opportunities to tighten this part. although this is not obligatory. The revised ms will be re-assessed by the referees.

**As suggested also from Rev #2, and in line with your comments, we further distribute section 3.4 between different subsections. In particular, previous sections 3.1, 3.2 and 3.3, were maintained within section 3, specifically focusing on the "Results". Subsection 3.4, was renamed as a separate section "4.Discussion and Comparison with other studies" and further distributed between three subsections: 4.1 Net Primary Production, Litterfall and Net Growth; 4.2. Net Ecosystem Production and Net Biomass Production; 4.3 Climate change scenarios, limitations and uncertainties.**

**Finally, we would highlight that, to improve the accessibility of colour figures for readers with colour vision deficiencies, the quality of all figures was tested through the Coblis-Color Blindness Simulator to check how the figures are perceived by readers with CVD.**

**Thank you for your positive review and for your constructive suggestions. Please find below our point-to-point answer to your comments:**

1. The exclusive focus of the analysis on fluxes, relegating the changes in stocks to the Supplement, may allow misinterpretation and misappropriation of the findings to justify further intensive management policies.

**The objective of our study is "to investigate the medium to long-term evolution of the forest C sink, as affected by the complex interactions between climatic variables and forest ecosystems", focusing on the methodological aspects. In this sense, the continuation of forest management (BAU) was chosen just to test our method, but this is not a policy scenario. Taking into account also the comments provided from Rev #2, this was further explicitly highlighted both within the abstract (L. 21-22,) and within the main text (L. 136-137, 228-229, 787).**

**Since the CBM model used within our modelling framework does conserve mass, the sum of the fluxes is equal to the sum of the stock changes. For this reason, it would not be strictly needed to address both these aspects in the main text, however, taking into account also the editor's suggestions, we moved from the supplementary information to the main text Fig 13S (now reported as Figure 4), highlighting the long-term evolution of the broadleaves and conifers living biomass stock estimated under the reference scenario. This figure was further discussed within the main text on L 364-366, L 571-573, 584-585, 713 – 714)**

2. This concern has two components – the change in stocks themselves under different scenarios, and the dynamics of heterotrophic respiration (Rh). I suggest moving figures 5S and 6S to the main body, and discussing the interaction of fluxes, pools and management all together.

**Figures 5S and 6S report the relative stock change applied to conifers and broadleaves respectively, as derived from the combination between climate simulations and LPJ-GUESS and used as input for CBM (for this reason this additional information was added as supplementary material), to calibrate the growth functions against climate change. Losses from fires are included in the DGVM simulations but not harvest. Both harvest and fires are included in the CBM simulations. The effect of management on C stocks is now reported, within the main text, in Figure 4 (see point 1 above), and further discussed on the main text (i.e., L 364-366, L 571-573, 584-585, 713 – 714). The trend and the absolute level of C stock reported on this figure for the RS do not diverge from the C stock determined under climate change scenarios.**
**We understand the point highlighted by the reviewer, however, since we did not consider different management scenarios (because we did not assess policy scenarios linked to various management strategies), we mostly focused our discussion on the fluxes. Nevertheless, as suggested also for the Editor, we added, as supplementary material, a specific figure reporting the evolution of Rh both under the reference scenario and climate change scenarios (see Fig. 8S). This figure was further discussed also within the main text (L 415, 438-439, 575 – 578, 696-697).**

3. While the short-term flux dynamics certainly will reflect the developmental stage they are currently in, the harvest intensity must be balanced with the long-term NEP. Maximizing NEP does not maximize the climate mitigation potential of forests.

**We recall again the fact that, within the present study, we did not aim to provide any policy scenario analysis, therefore we never stated that we should "maximize NEP". However, taking into account your suggestions, our discussion further highlighted that (i) a high NEP is an indication that the forest operates as a strong C sink, and (ii) to maximize the overall contribution of the forest sector to climate change mitigation, we need to maximize the "net sector productivity", including NEP and the net contribution of HWP emissions (see L. 649-655).**

4. It would be appropriate to acknowledge that the depiction of Rh in LPJ-GUESS does not reflect the latest understanding that Rh can be partly decoupled from NPP (https://doi.org/10.1029/2020GL092366), and that management-related disturbances can stimulate Rh for years to decades (https://doi.org/10.1016/j.foreco.2015.05.019; https://agupubs.onlinelibrary.wiley.com/doi/full/10.1029/2010JG001495; https://www.nature.com/articles/d41586-019-01026-8). These factors likely contribute to Rh being underestimated in the LPJ-GUESS simulations.

**In our study we used only net growth changes from LPJ-GUESS – as affected by climate change and fires – and we do not use Rh from that model in this analysis. In addition, we used "soc2005" LPG-GUESS simulations that uses fixed year-2005 land use and other human impacts. Whether or not LPJ-GUESS represents Rh properly does not at all affect the outcomes of our study. In fact, the main point of the reviewer – i.e. that disturbances, including harvesting, can affect Rh for years to decades - is well represented in the CBM-CFS3. That is why NEP changes over time across the scenarios. Moreover, Rh in CBM-CFS3 is temperature dependent (and the temperature is varying within our simulation) – and thus is can and does vary independent of NPP and it is not assumed to be a fixed proportion of NPP. Some of these points were also addressed within the discussion, on L 700 - 702**

5. I understand that a rigorous evaluation of these aspects is not feasible, but adding a paragraph to summarize remaining unknowns about soil C dynamics is appropriate, in my opinion. This section could also include references to the effect of nutrient availability (including deposition) on productivity, carbon allocation and the dynamics between plants and rhizosymbionts. There is growing evidence that these relationships are currently changing and may affect the growth and fitness of organisms involved, including changing the functional balance of soil microbial communities (leading to higher Rh).

**Thanks for your suggestion, we added a paragraph to mention remaining uncertainties in these models (see L. 737 - 740). However, we recall that, in this case, soil C dynamic is represented in the CBM. Other studies have previously assessed the uncertainty of these parameters, within the CBM (see for example, Smyth *et al., 2009; Hararuk* et al., 2017; Blujdea et al., 2021)**

6. While the use of wood in various products was not a factor in the current analysis, it may be appropriate to acknowledge that recent assessments of the substitution benefits of forest products conclude that these have likely been overestimated (http://dx.doi.org/10.1088/1748-9326/ab1e95, https://doi.org/10.1038/s41598-020-77527-8).

**We already highlighted within our conclusions, that "the additional mitigation potential provided from carbon storage in harvested wood products and material and energy substitution were not considered in our study and it is "unlikely that they would compensate for the**

reduction of the C sink directly provided by forest ecosystems" (L. 762-764). Taking into account the reviewer's suggestions, we added a further reference to Leturcq, 2020 (L. 764)

7. Finally, while the paper is overall well written and easy to follow, there are a number of typographical errors (duplication of words and punctuation marks, and minor grammatical errors) that are easy to fix using the spell checker.

**Many thanks for highlighting this point, all the text was carefully revised, taking into account also the specific comments provided from Rev #2.**

**Authors' replies to Reviewer #2 comments**
**Thank you for your positive review and for your useful and constructive suggestions. We will certainly take into considerations your comments when we revise the manuscript. Meantime, we take this opportunity to clarify some points.**

1. …. And here is my only criticism: in the paper the authors draw policy conclusions like the study "may constitute a first benchmark to set up specific management strategies". Due to the limitations of the modelling approach and the rather crude assumptions on the reference scenario conclusions on needed responses to revert the declining trend should not be drawn. As the authors mention in their response to Anonymous Referee #1 "the continuation of forest management (BAU) was chosen just to test our method, but this is not a policy scenario". In that sense the manuscript should more carefully draw conclusions on how to respond to the scenario results. Instead, the authors could provide requirements for making the results more policy relevant, e.g. by a more policy-oriented scenario design, sensitivity analyses regrading forest management options etc.

**We understand the point highlighted by the Reviewer and, taking also into consideration the comments provided from Rev#1, we recalled, both in the abstract (L 21-22) and in the main text (L , 136-137, 228-229, 785), the fact that our study does not aim to analyze a policy scenario. In particular, following your suggestion, we revised the last statement of our conclusions (see L. 790-794), highlighting that our methodological framework may help other studies to be more policy relevant. In this sense, further studies should certainly include, (i) a sensitivity analyses on different forest management options, and the consequent effects on the overall harvest levels, (ii) an assessment of the direct effect of these removals on the HWP net C sink and, possibly, (iii) even a first assessment of the possible indirect substitution benefits.**

2. Lines 205 ff: This explains… This is not clear to me: please more explicitly explain the geographical differentiation in the RC 2000-2015

**Figure 1 reports, on the upper panels, the geographical distribution of the average Net Growth estimated by CBM within the historical period 2000 – 2015 for broadleaved species, on the left side, and conifers, on the right side. Since the net growth represents the net biomass increment before losses from disturbances, these values are directly proportional to the Net Annual Increment (NAI) reported from NFI data. Despite the methodological differences between various European countries (Tomter et al., 2016), the NAI reported from Mediterranean countries and Northern European countries is generally lower than the NAI reported from central European countries (see for example Table 3.1-1, Lanz and Marchetti, 2020). Since the same NFI data are mostly used as input to initialize the CBM model, these differences – between**

**the NAI of various regions - explains the lower Net Growth generally estimated for Mediterranean and Northern European countries and the higher values estimated for Central European regions. This point was clarified within the revised version of the manuscript on L 285-291.**

3. Line 213 ff: This is… You attribute the reduction of NG "partially" to "ageing". I think this needs to be verified and better supported by data. Also the "rejuvenation" might reduce NG if it moves the biomass stock of a stand below the maximum increment. That there is instead a saturation effect can be observed from Figure 13S. I suggest adding a sentence on these dynamics and refer to this figure at this point as the biomass stock/increment relations and dynamics cannot be observed from Fig 1.

**We understand your point and, in line with your suggestions, we expanded the text on L. 308-309, also highlighting the saturation effect, as can be observed from figure 13S, now moved to the main text as Figure 4 (see also L. 364 - 368).**

4. Lines 245 - 280 Fig 1 - 2: The labels of what the panels show are very small. I suggest adding "upper panel right:…" etc. to the figure caption for better readability

**Thanks for your suggestion, we increased the size of all the labels and expanded the caption, both within the main text and in the article (see Fig. 1, 2, 5, 6, and supplement materials).**

5. Lines 289-291: At the European… The sentence is hard to understand. I suggest splitting it into two for more comprehensiveness.

**Thanks for your suggestion, the sentence was rephrased and split between two sentences (see L. 337-338).**

6. Section 3.4. "Comparison with other studies, limitations, and uncertainty of the present study" is very long. It addresses different aspects that should rather be separated for more readability and clearer structure. Suggestions for additional sections are: Comparison of the reference period with other data sources, Assessing impacts of climate change, Assessing underlying trends of growth dynamics, Limitations of the model, …

**Thanks for your suggestion, the overall section was further distributed between 3 different subparagraphs (4.1, 4.2, 4.3), specifically dedicated to the discussion and comparison with other studies (new Section 4).**

7. Lines 692-695: Due to the… The expected ranges for RCP 2.6 and 6.0 for 2050 are not clear from this sentence. Please revise it, e.g., by using the formulation in the abstract where it is much clearer.

**Thanks for your suggestion, the sentence was rephrased (L 771-772)**

**Additional technical corrections suggested from Rev #2:**

- Line 65: so-called **-> deleted**
- Line 79: combining **-> corrected (L. 133)**

- Caption Figure 1S: LP-GUESSJ should be LPJ-GUESS **-> corrected**
- Line 117: delete by year **-> deleted (L. 184)**
- Line 155-157: Sentence starting "This is…" unclear, please revise. **-> the sentence was revised, together with the previous one (L. 224-230)**
- Line 160: Suggest naming the output variables after "output" again (Growth Multipliers and area burnt) **→ the sentence was edit (L 231)**
- Line 568: in Figure 8 **-> corrected (L. 603)**
- 570-573: language, consider splitting sentence into two **-> the sentence was revised (L. 607-610)**